# Long-Term Nitrogen Addition Accelerates Litter Decomposition in a *Larix gmelinii* Forest

Miao Wang [1], Guancheng Liu [1], Yajuan Xing [1,2], Guoyong Yan [1] and Qinggui Wang [1,*]

1  School of Life Sciences, Qufu Normal University, 57 Jingxuan West Road, Qufu 273165, China;
   wangmiao0515@qfnu.edu.cn (M.W.); lxtxlgc123@126.com (G.L.); xingyajuan@hlju.edu.cn (Y.X.);
   guoyongyan1991@163.com (G.Y.)
2  Department of Agricultural Resource and Environment, Heilongjiang University, 74 Xuefu Road,
   Harbin 150080, China
*  Correspondence: qgwang1970@163.com; Tel.: +86-537-7038967

**Abstract:** Elevated atmospheric N deposition has the potential to alter litter decomposition patterns, influencing nutrient cycling and soil fertility in boreal forest ecosystems. In order to study the response mechanism of litter decomposition in *Larix gmelinii* forest to N deposition, we established four N addition treatments (0, 25, 50, 75 kg N ha$^{-1}$ yr$^{-1}$) in the Greater Khingan Mountains region. The results showed that (1) both needle and mixed leaf litter (*Betula platyphylla* and *Larix gmelinii*) exhibited distinct decomposition stages, with N addition accelerating decomposition for both litter types. The decomposition of high-quality (low C/N ratio) mixed leaf litter was faster than that of low-quality needle litter. (2) Mixed leaf litter increased the decomposition coefficients of litter with lower nutrients. (3) All N addition treatments promoted the decomposition of needle litter, while the decomposition rate of mixed leaf litter decreased under high-N treatment. (4) N addition inhibited the release of N and P in needle litter and promoted the release of N in mixed leaf litter, while high-N treatment had no positive effect on the release of C and P in mixed leaf litter. Our research findings suggest that limited nutrients in litter may be a key driving factor in regulating litter decomposition and emphasize the promoting effect of litter mixing and nitrogen addition on litter decomposition.

**Keywords:** nitrogen deposition; litter decomposition; stoichiometric ratios; boreal forest

## 1. Introduction

Human activities have doubled the annual input rate of inorganic nitrogen (N) into terrestrial ecosystems (43.47 Tg N yr$^{-1}$). It is anticipated that the increase in N input will continue to rise in the coming decades [1,2]. The augmented N deposition in certain regions has led to N overload in terrestrial ecosystems, exerting profound impacts on the carbon (C)-N cycles of the entire terrestrial ecosystem. Litter decomposition is a crucial process for nutrient cycling in terrestrial ecosystems [3–5] and represents a significant source of atmospheric CO$_2$ [6]. It provides available N and other essential elements for plant growth in terrestrial ecosystems [7]. The decomposition of litter is influenced by various biotic and abiotic factors. Globally and regionally, climate is recognized as the primary regulator of litter decomposition [8,9]. However, at the individual level, the quality of litter and the decomposition environment play crucial roles [10]. The decomposition of litter is primarily affected by the content of cellulose and lignin, as well as the stoichiometry of C:N. The quality of litter determines the degradability of organic matter and the availability of nutrients for decomposers [11,12]. The effectiveness of N in the external environment has long been a focus of attention regarding its impact on litter decomposition. Increased N deposition can alter the chemical composition of litter and modify the composition of soil decomposer communities and enzyme activities, thus changing the pattern of litter decomposition [13,14].

The deposition of N also has profound effects on the decomposition of litter. The addition of N can alter the quality of litter by promoting N uptake and influencing nutrient absorption in leaf, thereby indirectly affecting litter decomposition. It can also modify the decomposition environment and chemical composition of litter, thereby directly influencing the decomposition patterns of litter [13,14]. It is widely accepted that the decomposition of low-quality litter (high lignin content) is negatively impacted by long-term N addition, while the decomposition of high-quality litter (low lignin content) is enhanced by increasing N utilization [15,16]. Chen et al. have shown that the impact of N deposition on litter decomposition can be altered through external resource supply but ultimately depends on the internal stoichiometry of the decomposing litter [17]. N addition improves the quality of litter by increasing its N content, thereby reducing the carbon-to-nitrogen ratio. High-quality litter is preferentially decomposed by decomposers to obtain more nutrients. Knorr et al. further indicated via meta-analysis that litter decomposition was inhibited when the N addition was less than 75 kg N ha$^{-1}$ yr$^{-1}$ [18]. It is promoted when the N addition is in the range of 75–125 kg N ha$^{-1}$ yr$^{-1}$ but inhibited again when the N addition exceeds 125 kg N ha$^{-1}$ yr$^{-1}$. Tu et al. studied the decomposition processes of 10 forest litters and found that the decomposition rate of six forest leaf litters significantly decreased when the N addition exceeded 50 kg N ha$^{-1}$ yr$^{-1}$ [19]. The deposition of N is expected to alter the decomposition of woody debris and litter, but the intensity and direction of decomposition remain uncertain.

Under natural conditions, litter from various tree species commonly coexists on the forest floor, where different types of litter undergo decomposition collectively. There exists a notable contrast in decomposition rates between single-species leaf litter and mixed leaf litter. Within forest ecosystems, interactions among different leaf litter types exert intricate effects on the decomposition process. Nutrients released from rapidly decomposing litter may accelerate the decomposition of other litter, resulting in synergistic effects. Conversely, the breakdown of recalcitrant compounds released during decomposition may act antagonistically on litter decomposition [20,21]. The meta-analysis of Gartner and Cardon also noted 50% synergistic effects and 20% antagonistic effects in all experiments involving mixed leaf litter layers [22]. However, Wang et al. discovered that while N addition promoted the decomposition of *Pinus tabuliformis* litter in natural forests, it counteracted the suppressive effect on *Quercus mongolica* litter, thus having a neutral effect on the litter mixture [23]. These diverse findings hinder our comprehension of the decomposition mechanisms of woody debris under nitrogen-rich conditions.

The decomposition of leaf litter also depends on a series of mediated mechanisms involving soil animals and microorganisms [24–26]. A previous global meta-analysis found that the presence of soil fauna significantly increased the decomposition of terrestrial biomes' litter by an average of 37% [27]. However, research into the impact of soil fauna on the decomposition of woody litter under N deposition conditions is scarce, being primarily associated with the diversity, abundance, and activity of soil fauna [28]. In addition, research into the influence of soil animals on litter decomposition under N deposition is scarce. It has been reported that soil available N is the most crucial factor affecting mite abundance and soil animal communities [29]. Species composition and abundance can directly (through metabolism) and indirectly (by altering the habitat to enhance microstructural microbial activity) promote the decomposition and nutrient cycling of woody debris, altering not only the physical properties but also the chemical composition of organic matter. These mechanisms constitute the breakdown of plant litter structural components by extracellular enzymes and the recovery of organic N and phosphorus (P), forming a dynamic nutrient cycling system [30]. Differences in the chemical composition of litter can also affect the rate at which decomposers in the community break down litter [31]. Soil animals contribute only 5% to soil respiration, with their main impact considered to be enhancing microbial activity [32], thereby regulating the dynamics of litter carbon (C) and nutrients. Additionally, there are significant differences in the activity of different enzymes in different types of leaf litter, all of which can influence the response of litter

decomposition to N inputs. However, the relationship between soil animal communities and litter decomposition under long-term N addition is still unclear.

While the significance of litter decomposition in the biogeochemical cycles of ecosystems is widely acknowledged, the impact of N on this fundamental process remains uncertain. Concurrently, interactions between the litters of different tree species may alter the local decomposition environment, influencing the mode of decomposition. In China, boreal forests are distributed in the Greater Khingan Mountains region, considered to be one of the ecosystems most limited by N. Given that *Larix gmelinii* is the predominant tree species in our experimental plot, with a minor presence of *Betula platyphylla*, it remains unclear whether the introduction of *Betula platyphylla* litter would modify the decomposition dynamics of *Larix gmelinii* single litter. To enhance the effective protection and management of the surface litter in *Larix gmelinii* forests and deepen our understanding of the soil C cycle, we initiated a long-term N addition experiment in the Greater Khingan Mountains in 2011, and our objective was to investigate the impact of N addition on the decomposition of needle and mixed leaf litter in *Larix gmelinii* forests. This study aims to address the following questions: (1) Can N addition influence the decomposition of needle and mixed leaf litter? (2) Are the intensity and direction of N addition effects on litter decomposition in *Larix gmelinii* forests regulated by the N addition gradient? (3) Does the introduction of *Betula platyphylla* litter alter the decomposition dynamics of *Larix gmelinii* litter under increased N conditions? Our hypotheses posit the following predictions: (1) N addition accelerates litter decomposition, with the decomposition rate of mixed leaf litter surpassing that of needle litter. (2) N addition reduces the C:N ratio, thereby promoting the decomposition of both types of litter.

## 2. Materials and Methods

### 2.1. Site Description and Experimental Design

The experimental site is situated within the Nanwenghe National Natural Reserve in the Greater Khingan Mountains, Northeastern China (51°05′–51°39′ N, 125°07′–125°50′ E). The local climate typifies a cold temperate continental climate. The area experiences an annual average temperature of $-2.7\,°C$, with an annual maximum of $36\,°C$ and a minimum of $-48\,°C$. Sunshine duration amounts to 2500 h annually, with a plant growth period extending approximately 110 days. Annual precipitation averages around 500 mm, with the majority falling during the summer months of July and August, while annual evaporation stands at approximately 1000 mm. The predominant tree species is *Larix gmelinii*, with a stand density of $2852 \pm 99$/ha and an average Diameter at Breast Height (DBH) of $8.98 \pm 0.32$ cm.

To investigate the impact of N deposition, an N addition experiment was initiated in May 2011 (Figure 1). Twelve plots measuring 20 m × 20 m were established, with a 10 m wide buffer zone used to minimize inter-plot interference. Within each plot, five litter collectors (PVC pipe and gauze baskets, 1 m × 1 m × 1 m) were deployed, totaling 60 collectors. Based on current N deposition rates ($25\,kg\,N\,ha^{-1}\,yr^{-1}$) observed in Northern China [33], four treatment levels were implemented: control ($0\,kg\,N\,ha^{-1}\,yr^{-1}$), low nitrogen (LN, $25\,kg\,N\,ha^{-1}\,yr^{-1}$), medium nitrogen (MN, $50\,kg\,N\,ha^{-1}\,yr^{-1}$), and high nitrogen (HN, $75\,kg\,N\,ha^{-1}\,yr^{-1}$). Each treatment was replicated three times to simulate anticipated atmospheric N deposition trends over 1, 2, and 3 times the future nitrogen deposition rate. N application commenced in May 2011 and was evenly distributed over five applications during the local forest's growing season (May to September). Prior to each application, $NH_4NO_3$ was measured according to the designated N addition rate, mixed with 32 L of water, and uniformly sprayed across the forested area of each plot. To maintain consistent water supply, the control plot received an equivalent amount of pure water during each application.

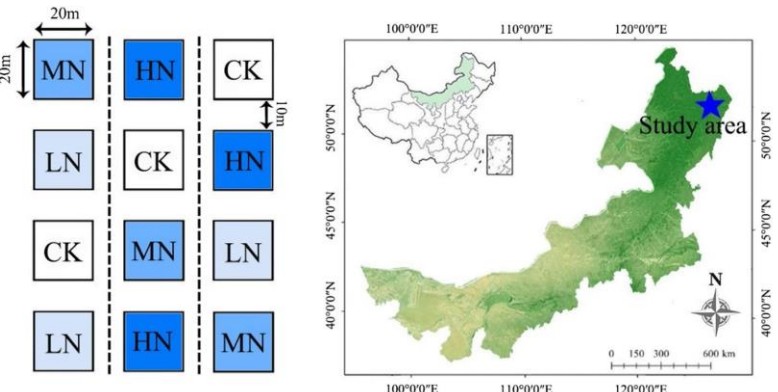

**Figure 1.** Nitrogen sampling plot and experimental design diagram within the forest ecosystem of the Greater Khingan Mountains in China. CK, LN, MN, and HN represent control and low-nitrogen treatment, medium-nitrogen treatment, and high-nitrogen treatment, respectively.

### 2.2. Sample Collection and Analysis

In September 2017, the litters in various litter collectors were collected. The litter samples were collected and dried at 60 °C for 48 h. Litters of *Larix gmelinii*, *Betula platyphylla*, and *Larix gmelinii* were mixed at a ratio of 1:1, each weighing 5 g, and packed in a 10 cm × 10 cm nylon net bag (pore diameter is 1 mm). In October 2017, the litter bags were placed on the forest floor (each litter bag is 2 cm apart) to represent the natural fallen leaf. A total of 432 litter bags were placed. Since the experimental area was snowed on heavily from November to April, the litter was collected in May (spring), July (summer), and September (autumn) of 2018 and 2019 according to the seasonal dynamics. A bag of mixed leaf litter bag and a bag of needle litter bag were randomly selected from each plot a total of 6 times, with 72 bags in total for one time. The dry weight of the litter was measured to determine the mass loss. After grinding, we determined the litter total C(TC), total N(TN), and total P (TP).

Each time the litter bags were collected, soil columns with heights of 15 cm and widths of 20 cm were dug out. Small soil animals were collected via the improved Tullgren funnel method [34]. They were separated under a 25 W incandescent lamp and observed every 8 h for a total of 24 h. Finally, all the extracted animal samples were stored in 75% ethanol and identified at the family level or suborder level in sequence and then counted under stereomicroscope (OLYMPUS SZX16, Japan).

The total N content of litter and soil were determined using a continuous flow analyzer (SKALAR SAN+, The Netherlands). Total C was determined using a automatic TC/TN analyzer (Analytik Jena AG, multi N/C 3100, Germany). Total P was determined via molybdenum antimony anti-colorimetry and an ultraviolet visible spectrophotometer (SHIMADZU UV-1780, Japan). The deionized water was added to 4 g air dried soil (soil/water ratio of 1:2.5) and remained silent for half an hour, and then we measured the soil pH with a pH meter (Sartorius PB-10, Gottingen, Germany).

### 2.3. Data Analysis

In order to describe the residual processes of nutrients in litter decomposition, a widely used model for describing mass loss data was selected: a single exponential model [35]. All statistical analyses were performed using SPSS v22.0 (IBM SPSS Statistics, Inc., Armonk, NY, USA), and graphs were drawn using Sigmaplot 12.5 (Systat Software Inc., Chicago, IL, USA).

Olson's negative exponential fitting equation was defined as follows:

$$Y = a\,e^{-kt} \tag{1}$$

where t is the decomposition time (year), k is the decomposition coefficient of litter $(kg \cdot kg^{-1} \cdot yr^{-1})$, and Et is the residual percentage of organic matter or element content in litter leaf at t time.

$$E_t (\%) = [(M_t \times C_t)/(M_o \times C_o)] \times 100\% \tag{2}$$

where $M_t$ represents the litter quality (g) in t time and $M_o$ represents the initial litter quality (g). $C_t$ represents the nutrient content of litter leaf in t time (g/kg; organic matter: %). $M_o$ represents the nutrient content of the initial litter (g/kg; organic matter: %).

The time required for 50% decomposition was as follows:

$$T_{0.5} = \ln 0.5^{(-K)} \tag{3}$$

The time required for 95% decomposition was as follows:

$$T_{0.95} = \ln 0.05^{(-K)} \tag{4}$$

We assessed the normal distribution of all data using the Kolmogorov–Smirnov test and tested the homogeneity of variances with Levene's test. One-way analysis of variance (ANOVA) was employed to investigate the effects of N addition on the stoichiometric ratios of needle and mixed leaf litter. Additionally, a general linear model multivariate analysis of variance (ANOVA) was utilized to examine the effects of time, treatments, and their interactions on the stoichiometric characteristics of litter. Regression analysis was applied to explore the relationship between chemometrics and the mass residues of needle and mixed leaf litter. To assess differences between treatments, Tukey's post hoc test was conducted, with statistically significant differences accepted at $p < 0.05$. Furthermore, redundancy analysis (RDA) was performed to evaluate the relationship between litter decomposition, soil fauna, and soil properties. To elucidate the main factors affecting the decomposition of litter quality, a random forest model was constructed using the random forest function from the random forest package in R-4.2.1. Data analysis and graphical visualization were conducted using the R platform (version 4.2.1, R Core Team, 2018).

## 3. Results

### 3.1. Soil and Litter Property

N addition had no significant effect on the soil TC and TP, but MN and HN treatments significantly increased the soil TN. HN treatment significantly increased the soil N:P and significantly reduced the soil pH. The TC, TN, and TP of mixed leaf litter was higher than that of needle litter. Moreover, the ratio of C:N of mixed leaf litter was also lower than that of needle litter, and the initial quality of mixed leaf litter was higher (Tables 1 and 2). HN significantly increased TN content of mixed leaf litter and needle litter (Table 1). Overall, N addition promotes the release of C and N from litter, while inhibiting the release of P (Figure 2). We found that the C:N of needle litter decreased with the decomposition time, while the C:N of mixed leaf litter decreased in the early stage and increased in the later stage. HN significantly reduced the C:N of needle litter but had no significant effect on mixed leaf litter (Figure 3C,D).

### 3.2. Effects of N Addition on Decomposition of Needle and Mixed Leaf Litter

In the two years of decomposition after N addition, the mass loss patterns of both needle and mixed leaf litter showed continuous decline (Figure 2A,E). Based on the single exponential model, we found that N had a significant effect on the litter decomposition rate. N addition increased the decomposition coefficients of needle litter by 22%–46% and mixed leaf litter by 1%–29%. Within the first year (2018), needle litter lost about 20% of its original quality. The order of decomposition coefficient (k) of needle litter under different N treatments was HN > MN > LN > CK (Table 3). The order of decomposition coefficient of mixed leaf litter was MN > LN > HN > CK. The needle litter decomposition values for LN

and MN treatments were shortened by 0.79 years and 1.03 years, respectively, while that of HN treatment shortened by 1.4 years. LN, MN, and HN treatments shortened mixed leaf litter decomposition by 0.1, 0.88, and 0.04 years, respectively. If 95% of the needle litter was decomposed, the decomposition times of LN and MN treatments shortened by 3.43 years and 4.44 years, respectively, and that of HN treatment shortened by 6.06 years. If 95% of the mixed leaf litter was decomposed, the decomposition times of LN and MN treatments could be shortened by 0.47 years and 3.82 years, respectively, and that of HN treatment could be shortened by 0.19 year (Table 3). Different N treatments promoted the decomposition of needle litter, and the effect was reinforced with the increase in N addition. Mixed leaf litter lost more mass than needle litter (Figure 2A,E), but HN treatment inhibited the decomposition of mixed leaf litter (Table 3).

**Table 1.** Initial element and some organic matter content (collected from litter collectors) of needles and mixed leaf litter (mean $\pm$ SD).

| Treatments | CK | LN | MN | HN |
|---|---|---|---|---|
| Needle litter C (g·kg$^{-1}$) | 493.46 $\pm$ 32.83 [a] | 491.30 $\pm$ 16.44 [a] | 486.46 $\pm$ 35.78 [a] | 488.96 $\pm$ 13.29 [a] |
| Needle litter N (g·kg$^{-1}$) | 3.22 $\pm$ 0.11 [b] | 3.24 $\pm$ 0.11 [b] | 3.35 $\pm$ 0.11 [b] | 3.71 $\pm$ 0.10 [a] |
| Needle litter P (g·kg$^{-1}$) | 0.99 $\pm$ 0.08 [a] | 0.90 $\pm$ 0.05 [ab] | 0.83 $\pm$ 0.04 [ab] | 0.80 $\pm$ 0.01 [b] |
| Needle litter C:N | 153.43 $\pm$ 15.52 [a] | 151.54 $\pm$ 1.71 [a] | 145.29 $\pm$ 10.80 [ab] | 131.79 $\pm$ 0.23 [b] |
| Mixed leaf litter C(g·kg$^{-1}$) | 515.11 $\pm$ 42.97 [a] | 528.92 $\pm$ 8.83 [a] | 532.08 $\pm$ 2.67 [a] | 524.52 $\pm$ 9.67 [a] |
| Mixed leaf litter N (g·kg$^{-1}$) | 3.64 $\pm$ 0.05 [c] | 4.38 $\pm$ 0.03 [b] | 4.50 $\pm$ 0.04 [b] | 4.80 $\pm$ 0.09 [a] |
| Mixed leaf litter P (g·kg$^{-1}$) | 1.14 $\pm$ 0.01 [a] | 1.11 $\pm$ 0.04 [ab] | 1.09 $\pm$ 0.05 [ab] | 1.03 $\pm$ 0.02 [b] |
| Mixed leaf litter C:N | 141.58 $\pm$ 10.22 [a] | 120.82 $\pm$ 1.25 [b] | 118.24 $\pm$ 0.70 [b] | 109.36 $\pm$ 3.74 [b] |

CK, LN, MN, and HN represent the control and low-nitrogen, medium-nitrogen, and high-nitrogen addition treatment, respectively; different superscript letters within each column represent significant differences between treatments ($p < 0.05$).

**Table 2.** General characteristics of soil in the field.

| Year | Treatment | T C (g kg$^{-1}$) | T N (g kg$^{-1}$) | T P (g kg$^{-1}$) | C:N | C:P | N:P | pH |
|---|---|---|---|---|---|---|---|---|
| 2017 | CK | 43.24 $\pm$ 5.52 [a] | 1.76 $\pm$ 0.09 [b] | 0.71 $\pm$ 0.06 [ab] | 24.75 $\pm$ 4.17 [a] | 60.74 $\pm$ 2.79 | 2.51 $\pm$ 0.34 [b] | 5.48 $\pm$ 0.14 [a] |
| | LN | 49.69 $\pm$ 3.47 [a] | 1.96 $\pm$ 0.19 [ab] | 0.79 $\pm$ 0.03 [a] | 25.77 $\pm$ 3.96 [a] | 62.67 $\pm$ 4.09 | 2.48 $\pm$ 0.33 [b] | 5.38 $\pm$ 0.13 [a] |
| | MN | 47.64 $\pm$ 5.23 [a] | 2.30 $\pm$ 0.09 [a] | 0.70 $\pm$ 0.01 [ab] | 20.67 $\pm$ 1.69 [a] | 67.70 $\pm$ 7.92 | 3.27 $\pm$ 0.14 [ab] | 5.23 $\pm$ 0.05 [ab] |
| | HN | 49.01 $\pm$ 2.79 [a] | 2.26 $\pm$ 0.07 [a] | 0.65 $\pm$ 0.03 [b] | 21.64 $\pm$ 0.72 [a] | 75.82 $\pm$ 6.44 | 3.50 $\pm$ 0.18 [a] | 5.01 $\pm$ 0.02 [b] |
| 2018 | CK | 45.81 $\pm$ 6.89 [a] | 2.00 $\pm$ 0.32 [c] | 0.83 $\pm$ 0.14 [a] | 22.52 $\pm$ 1.08 [a] | 52.48 $\pm$ 2.6 [a] | 2.41 $\pm$ 0.03 [b] | 5.52 $\pm$ 0.12 [a] |
| | LN | 45.25 $\pm$ 2.36 [a] | 2.09 $\pm$ 0.17 [bc] | 0.79 $\pm$ 0.19 [a] | 21.81 $\pm$ 0.36 [a] | 56.96 $\pm$ 4.74 [a] | 2.77 $\pm$ 0.31 [b] | 5.41 $\pm$ 0.23 [ab] |
| | MN | 45.80 $\pm$ 3.73 [a] | 2.54 $\pm$ 0.22 [ab] | 0.81 $\pm$ 0.17 [a] | 18.62 $\pm$ 0.31 [b] | 57.55 $\pm$ 5.00 [a] | 3.30 $\pm$ 0.17 [a] | 5.29 $\pm$ 0.06 [ab] |
| | HN | 48.61 $\pm$ 5.72 [a] | 2.60 $\pm$ 0.27 [a] | 0.79 $\pm$ 0.11 [a] | 18.36 $\pm$ 0.87 [b] | 62.29 $\pm$ 3.07 [a] | 3.32 $\pm$ 0.21 [a] | 4.98 $\pm$ 0.05 [b] |

CK, LN, MN, and HN represent the control and low-nitrogen, medium-nitrogen, and high-nitrogen addition treatments, respectively; TC, TN, and TP represent soil total carbon, soil total nitrogen, and soil total phosphorous, respectively; C:N, C:P, and N:P represent the carbon–nitrogen ratio, carbon–phosphorous ratio, and nitrogen–phosphorous ratio. Different superscript letters within each column represent significant differences between treatments ($p < 0.05$).

### 3.3. Relationship of Decomposition between Soil Animals and the Needle or Mixed Leaf Litter under N Addition

Decomposed time and treatments significantly impacted the C content of mixed leaf litter and the C:N ratio of needle litter (Table 4). The number of soil fauna significantly increased for the LN and MN treatments, while it significantly decreased for the HN treatment (Table 5). A significant positive correlation was observed between the C/N ratio and needle litter mass loss, which was enhanced by N addition. Notably, under LN treatment, this correlation was the strongest. Conversely, under HN treatment, the C/N ratio of mixed leaf litter exhibited a negative correlation with the mass loss (Figure 3A,B). The decomposition time, treatments, and their interactions significantly influenced the C, N, and P contents of needle litter, as well as the N, P, and C:N ratios of mixed leaf litter. A random forest model was employed to predict litter quality decomposition. It revealed that the C, N, and P content of litter, as well as soil N, significantly influenced the decomposition of needle litter, whereas litter and soil P had a significant impact on the

decomposition of mixed leaf litter (Figure 4). The first and second RDA axes explained 92.51% and 2.21% of the variance in the relationships between litter decomposition, soil fauna, and soil properties. Notably, Collembola and Oribatidae had a significant effect on litter decomposition (Figure 5).

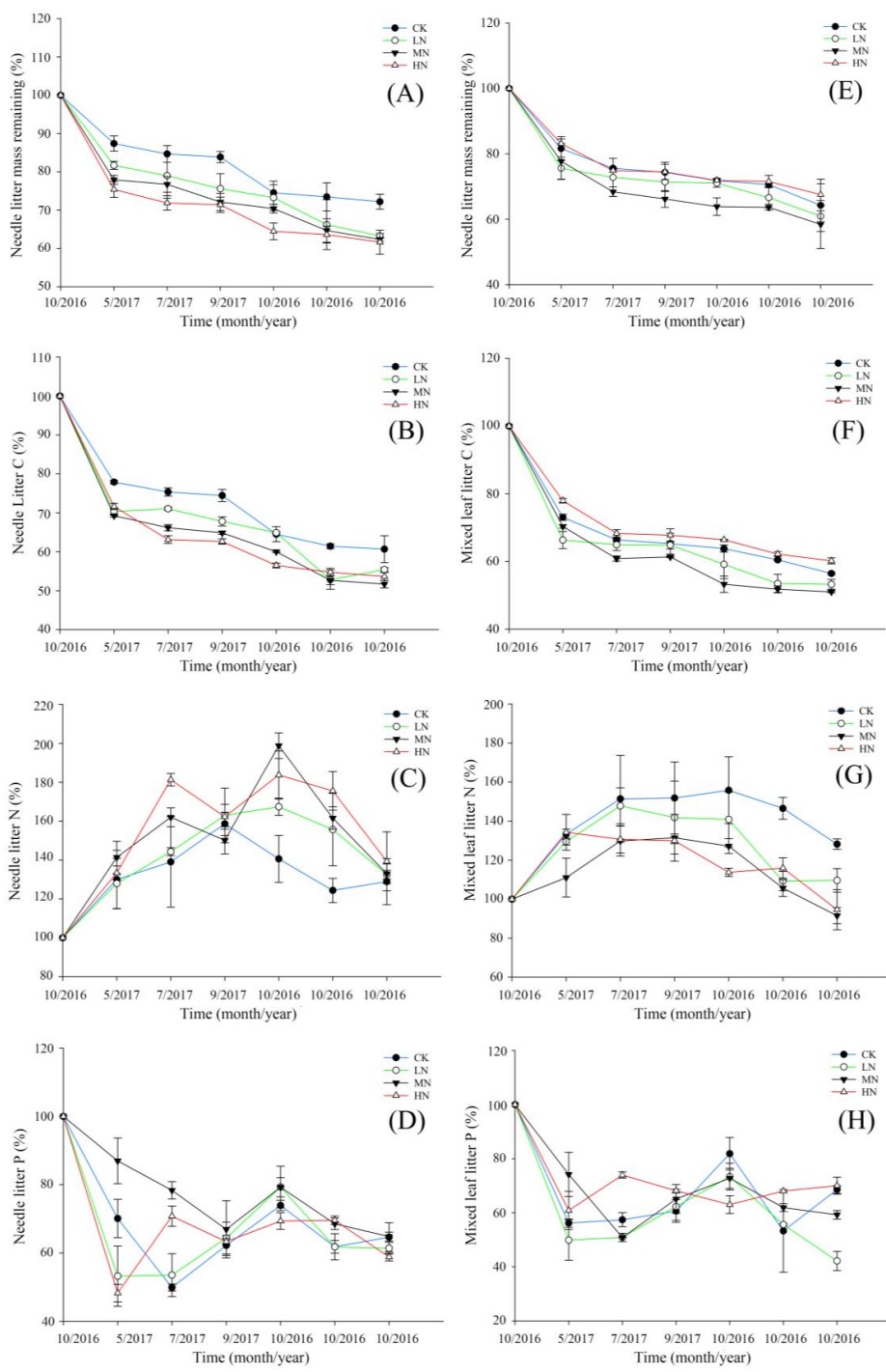

**Figure 2.** Mass and nutrient remains of needle and mixed leaf litter under different nitrogen treatments. CK, LN, MN, and HN represent the control and low-nitrogen, medium-nitrogen, and high-nitrogen addition treatments, respectively. (**A**) The needle litter mass remaining; (**B**) the needle litter C content (%); (**C**) the needle litter N content (%); (**D**) the needle litter P content (%); (**E**) the mixed leaf litter mass remaining; (**F**) the mixed leaf litter C content (%); (**G**) the mixed leaf litter N content (%); (**H**) the mixed leaf litter P content (%).

**Table 3.** Exponential fitting equations of litter residues of needle and mixed leaf litter under different nitrogen treatments.

| Treatments | a | Decomposition Coefficient (k) | Determinant Coefficients (R$^2$) | Time Required to Decompose 50% | Time Required to Decompose 95% |
|---|---|---|---|---|---|
| Needle litter CK | 0.987 | 0.156 ± 0.010 | 0.979 | 4.44 | 19.20 |
| Needle litter LN | 0.967 | 0.190 ± 0.025 | 0.921 | 3.65 | 15.77 |
| Needle litter MN | 0.933 | 0.203 ± 0.029 | 0.908 | 3.41 | 14.76 |
| Needle litter HN | 0.915 | 0.228 ± 0.034 | 0.898 | 3.04 | 13.14 |
| Mixed leaf litter CK | 0.938 | 0.176 ± 0.033 | 0.852 | 3.93 | 17.02 |
| Mixed leaf litter LN | 0.925 | 0.181 ± 0.033 | 0.861 | 3.83 | 16.55 |
| Mixed leaf litter MN | 0.919 | 0.227 ± 0.042 | 0.853 | 3.05 | 13.20 |
| Mixed leaf litter HN | 0.935 | 0.178 ± 0.032 | 0.860 | 3.89 | 16.83 |

a is the regression coefficients; CK, LN, MN, and HN represent the control and low-nitrogen, medium-nitrogen, and high-nitrogen addition treatments, respectively.

**Table 4.** Summary of a two−way ANOVA showing the effects of needle litter, mixed leaf litter stoichiometry and time, and treatments (CK, LN, MN, HN).

| Factors | F($p$) Value | | |
|---|---|---|---|
| | Time | Treatments | Time × Treatments |
| Needle litter C | 271.704 (<0.001) | 126.478 (<0.001) | 5.808 (<0.001) |
| Needle litter N | 17.248 (<0.001) | 13.904 (<0.001) | 2.793 (<0.001) |
| Needle litter P | 10.612 (<0.001) | 21.119 (<0.001) | 8.396 (<0.001) |
| Mixed leaf litter C | 9.843 (<0.001) | 3.250 (<0.05) | 1.597 (>0.05) |
| Mixed leaf litter N | 12.396 (<0.001) | 11.158 (<0.001) | 2.949 (<0.05) |
| Mixed leaf litter P | 9.171 (<0.001) | 11.459 (<0.001) | 5.706 (<0.001) |
| Needle litter C:N | 25.949 (<0.001) | 90.288 (<0.001) | 1.505 (>0.05) |
| Mixed leaf litter C:N | 15.628 (<0.001) | 8.848 (<0.001) | 3.126 (<0.001) |

CK represents the control; LN represents the low-N treatment; MN represents the medium-N treatment; HN represents the high-N treatment.

**Table 5.** Composition of the main soil fauna community under different treatments across all sampling periods.

| Month | Treatment | Isotomidae | Onychiuridae | Entomobryidae | Hypogastruridae | Oribatida | Mesostigmata |
|---|---|---|---|---|---|---|---|
| 5 | CK | 35 ± 5.31 [b] | 6 ± 1.25 [a] | 20 ± 5.72 [a] | 13 ± 6.13 [a] | 42 ± 3.26 [b] | 9 ± 1.25 [b] |
| | LN | 67 ± 13.47 [a] | 4 ± 1.25 [a] | 10 ± 2.45 [ab] | 10 ± 2.45 [a] | 57 ± 2.87 [a] | 25 ± 0.47 [a] |
| | MN | 21 ± 2.05 [bc] | 4 ± 1.63 [a] | 15 ± 4.90 [ab] | 14 ± 7.35 [a] | 33 ± 3.68 [c] | 8 ± 2.87 [b] |
| | HN | 10 ± 4.08 [c] | 5 ± 2.05 [a] | 6 ± 0.47 [b] | 17 ± 0.47 [a] | 32 ± 3.27 [c] | 9 ± 1.63 [b] |
| 7 | CK | 105 ± 4.08 [b] | 12 ± 0.47 [b] | 13 ± 4.50 [c] | 9 ± 0.00 [ab] | 75 ± 5.72 [b] | 19 ± 3.67 [a] |
| | LN | 201 ± 9.39 [a] | 26 ± 0.82 [a] | 61 ± 3.27 [a] | 16 ± 5.31 [a] | 174 ± 5.31 [a] | 12 ± 2.05 [ab] |
| | MN | 223 ± 18.78 [a] | 17 ± 2.45 [b] | 38 ± 8.98 [b] | 15 ± 0.00 [a] | 57 ± 10.61 [ab] | 7 ± 0.82 [bc] |
| | HN | 57 ± 3.27 [c] | 7 ± 0.82 [c] | 15 ± 0.47 [c] | 6 ± 0.82 [b] | 44 ± 8.98 [c] | 4 ± 0.00 [c] |
| 9 | CK | 37 ± 6.53 [a] | 4 ± 0.00 [a] | 7 ± 1.63 [b] | 6 ± 2.05 [a] | 39 ± 15.11 [a] | 9 ± 0.47 [a] |
| | LN | 17 ± 3.27 [b] | 5 ± 0.47 [a] | 12 ± 1.25 [a] | 5 ± 0.82 [a] | 19 ± 1.63 [ab] | 6 ± 2.05 [a] |
| | MN | 3 ± 0.82 [c] | 3 ± 0.00 [a] | 3 ± 0.00 [c] | 3 ± 0.82 [a] | 25 ± 11.8 [ab] | 6 ± 2.05 [a] |
| | HN | 12 ± 2.87 [bc] | 5 ± 1.63 [a] | 7 ± 0.00 [b] | 6 ± 1.63 [a] | 9 ± 0.81 [b] | 5 ± 1.63 [a] |

CK represents the control; LN represents the low-N treatment; MN represents the medium-N treatment; HN represents the high-N treatment. Different superscript letters within each column represent significant differences between treatments ($p < 0.05$).

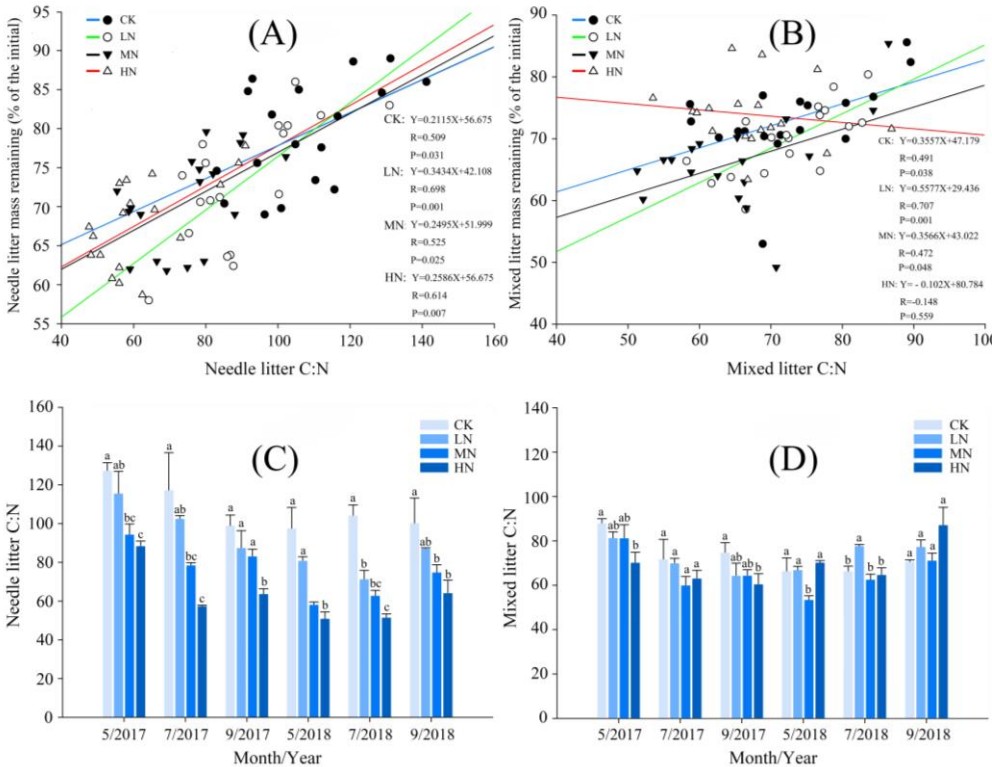

**Figure 3.** The relationship between the litter ratio of C:N and mass remaining under different nitrogen treatments. CK, LN, MN, and HN represent the control and low−nitrogen, medium−nitrogen, and high−nitrogen addition treatments, respectively. (**A**) The relationship between the needle litter ratio of C:N and needle litter mass remaining; (**B**) the relationship between the mixed leaf litter ratio of C:N and mixed leaf litter mass remaining; (**C**) the needle litter ratio of C:N; (**D**) the mixed leaf litter ratio of C:N. Different superscript letters within each column represent significant differences between treatments ($p < 0.05$).

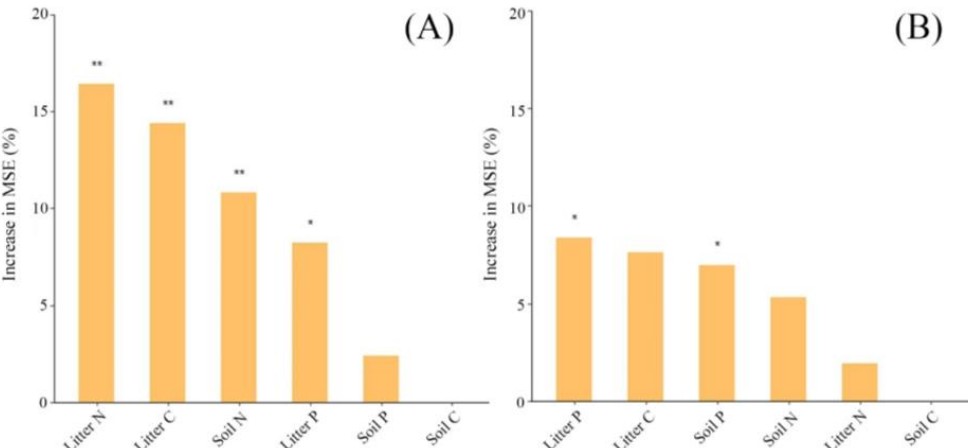

**Figure 4.** Random forest variable importance plot. The variables are ranked in order of relevance for predicting the decomposition of litter quality ((**A**) needle litter, (**B**) mixed leaf litter). The importance measure considered for the analysis is the mean decrease in accuracy computed via a random forest classification algorithm. MSE represents the mean square error, * represents $p < 0.05$, and ** represents $p < 0.01$.

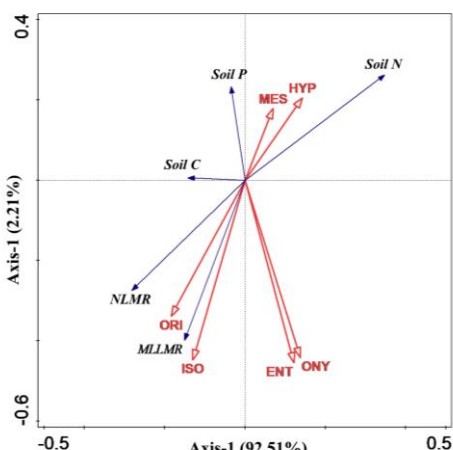

**Figure 5.** Perform redundancy analysis (RDA) evaluate the relationship between litter decomposition, soil fauna, and soil properties. NLMR, needle litter mass remaining; MLLMR, mixed leaf litter mass remaining; ISO, ONY, ENT, HYP, ORI, and MES represent Isotomidae, Onychiuridae, Entomobryidae, hypogastruridae, Oribatida, and Mesostigmata, respectively.

## 4. Discussion

### 4.1. Effect of N Addition on Decomposition Rate of Needle and Mixed Leaf Litter

Numerous studies have highlighted the variability in the impact of N addition on decomposition rates over different decomposition stages [36–38]. In our study, needle litter displayed decomposition rates of 15.1% in the first year and 13.9% in the second year. In contrast, mixed leaf litter exhibited higher decomposition rates of 27% in the first year and 7.1% in the second year. These decomposition patterns align with distinct stages observed in various studies [39]. Given that N strongly influences the early stages of decomposition, an increase in N concentration results in decreased litter C/N, elevated soluble matter, and cellulose decomposition, often leading to a stimulation of the overall decomposition rate. However, in later stages, the accumulation of lignin in decomposed litter provides physical protection to cellulose and other substances, potentially impeding the decomposition rate. Multiple studies have indicated the existence of a threshold for litter decomposition. Knorr et al. indicates that N addition within the range of 75–125 kg N ha$^{-1}$ yr$^{-1}$ enhances litter decomposition, but below or above this critical range, it inhibits decomposition [20]. N addition at rates of 25, 50, and 75 kg N ha$^{-1}$ yr$^{-1}$ indirectly stimulated litter mass loss post-decomposition, ranging from one to three times the rate of environmental N deposition. This observation aligns with evidence supporting increased litter mass loss due to N deposition. As hypothesized, the mass loss of mixed leaf litter surpassed that of needle litter throughout the experiment. This accelerated decomposition of mixed leaf litter can be attributed to nitrogen's pronounced influence during the early decomposition stages, affecting the physiological adaptation of decomposition-related organisms. Litter with high-N content (low C/N ratio) decomposes more rapidly than low-N content [40].

Our study revealed that the mixing of litter from two plant species increased the decomposition constant of lower-nutrient litter, exhibiting a synergistic effect (Table 2), potentially linked to the exchange in limiting nutrients between litters [41,42]. The initial quality of mixed leaf litter is higher than that of needle litter, and nutrients are transferred from high-quality litter to low-quality litter, thereby increasing the attraction of the species-specific microbiome. In species-rich plant communities, substantial differences in chemical composition between litters may lead to varied decomposition rates in mixed litter due to microhabitat interactions [24]. The significant morphological disparities between needles and broadleaved species, along with differences in physical properties, could influence the microclimate of litter decomposition by enhancing ventilation conditions while facilitating nutrient transfer. Net nutrient transfer between leaves in litter mixtures tends to favor low-nutrient species. Furthermore, increased soil animal activity and diversity in mixed leaf litter may contribute to this phenomenon, as observed in the research of Kaneko and

Salamanca [43], which discovered higher soil fauna abundance in mixed leaf litter compared to single-species litter, potentially providing richer nutrient resources for microorganisms and soil animals, thereby augmenting microbial diversity and mineralization.

### 4.2. Effects of N Addition and Small Soil Animals on Decomposition of Needle and Mixed Leaf Litter

The decomposition of mixed leaf litter was hindered under high-N treatment, likely attributed to the suppression of lignin enzyme synthesis induced by elevated N concentrations [44]. Wu et al. showed that soil pH serves as a reliable predictor of soil bacterial diversity across various terrestrial ecosystems [45]. Our results reveal a significant decrease in soil pH following N addition (Table 2), contributing to heightened soil bacterial diversity. Although this study did not explore potential changes in microbial community responses to N, observations of small soil animals indicated noteworthy reductions in the populations of Collembola and Oribatidae under high-N treatment (Table 3, Figure 5). Oribatid mites, recognized as "k" strategists, primarily feed on resilient wood matrices [46], while Collembola, identified as "r" strategists, typically prey on fungi and rely on unstable C [47]. These soil animals play a crucial role in regulating microbial community richness, diversity, activity, and diffusion, thereby influencing litter and biomass [48]. Aupic-Samain et al. emphasized significantly higher fungal biomass in coniferous forests compared to broad-leaved forests [49]. Considering the effect of litter types, we hypothesize that Collembola and Oribatidae preferentially participate in the decomposition of needle litter, given its greater resistance to degradation compared to broad-leaved litter.

Throughout the entire decomposition period, the high-N treatment suppressed the activities of soil animals, potentially explaining the lower decomposition rate of mixed leaf litter compared to needle litter under high-N treatment. Simultaneously, the C:N ratio of needle litter was significantly reduced by the high-N treatment, while the C:N ratio of mixed leaf litter was only positively influenced by the high-N treatment in the initial stages of decomposition. However, in the later stages of decomposition, the high-N treatment had no significant effect on the C:N ratio of mixed leaf litter, and the C:N ratio of mixed leaf litter under high-N treatment exceeded that of needle litter (Figure 3C,D). The observation of nutrient release from litter revealed that the C release of mixed leaf litter under high-N treatment remained unaffected by N addition. Additionally, the N release pattern of mixed leaf litter differed from the enrichment-release mode observed in needle litter, demonstrating a direct release (Figure 2). This discrepancy was one of the reasons why the C:N ratio of mixed leaf litter did not decrease in the later stages of decomposition under high-N treatment. These findings suggest that the decomposition mechanism of mixed leaf litter differs significantly from that of needle litter.

The results of the random forest model also indicate that, compared to needle litter, litter P and soil P are the primary factors influencing the quality decomposition of mixed litter (Figure 4). The P content in mixed litter is significantly higher than that in needle litter (Table 1). Zheng et al. found that litter with higher P (lower C/P and N/P ratios) decomposes more quickly [50]. The nutrient dynamics of litter can represent the nutrient availability of decomposer communities [51]. Nitrogen addition typically leads to P limitation, and previous studies have also found that the study site is in a state of P limitation [52]. The higher nutrient supply in mixed litter implies increases in both the quantity (more metabolic forms) and quality (easily mineralized through leaching) of the substrate. This will satisfy the decomposers' need for P, enhance soil microbial activity, and promote the loss of litter quality. Therefore, the limiting nutrient in litter may be a key driver regulating litter decomposition.

### 4.3. Effects of N Addition on Nutrient Release from Needle and Mixed Leaf Litter

N addition induced a significant alteration in the relationship between the C/N ratio and litter mass loss (Figure 3A,B). As the litter mass plays a crucial role in controlling the litter decay rate, the changes prompted by N may have a more pronounced influence on litter

decay rates, particularly in areas with high N deposition. N application notably enhanced C release (Figure 2B). Fresh litter typically provides ample C to meet the requirements of microbial decomposers, which need higher nutrient concentrations for biomass construction and maintenance. As litter decomposition rates increase, the elevated utilization rate may contribute to enhanced forest productivity and increased litter input. Consequently, heightened litter input leads to a rise in total C in forest land, thereby augmenting C pools in the soil. P release and enrichment are also influenced by biological factors, easily immobilized by microorganisms [53]. N application generally inhibits P release, resulting in a higher N return rate than P and diminishing soil P availability (Table 2). Under the natural state, the N content in litter initially increased and then decreased, achieving net N enrichment. Our findings reveal that N undergoes immobilization in the early stages of litter decomposition, especially in the initial year, followed by mineralization. It is widely accepted that N accumulation in litter primarily results from microbial immobilization. He et al. traced [15]N and discovered that a substantial portion of fixed N in the soil exists in the form of insoluble components in microbial tissues, such as fungal mycelia [54]. During the process of litter decomposition, the N level in microbial protoplasts on leaf litter increased over time. The application of a specific N concentration stimulated microbial activity, leading to increased exogenous N acquisition. With a rising N concentration, the level of N acquisition increased, resulting in N enrichment in litter. Consequently, N addition exhibited a tendency to impede N release from litter. Moreover, the sequence of N enrichment and release during litter decomposition varies under different N applications, likely due to differences in litter substrate mass, resulting in distinct N retention and mineralization times in different leaf types. Some studies propose that higher N content in litter corresponds to shorter retention times. Simultaneously, N application alters microbial N acquisition rates and the time required to reach the critical N threshold [55,56]. In contrast to the impact of N deposition on N release from needle litter, N application promoted N release from mixed leaf litter as a whole. This could be attributed to the nutrient-rich background of mixed leaf litter and the abundant N element, acting as the microbial nutrient base. Although there is still immobilization and mineralization, the overall decomposition of mixed leaf litter may not necessitate external N sources for mineralization, and the overall concentration level remains low.

In this study, we found that soil animals have a positive impact on the decomposition of litter. Soil animals can directly promote litter decomposition through fragmentation and feeding, as well as indirectly promote litter decomposition by changing the soil environment and interacting with microbial communities. In the future, by distinguishing different litter bag meshes, we will explore how different soil animal densities and diversity levels affect litter decomposition and the mechanisms of their interactions with microorganisms.

**5. Conclusions**

Needle and mixed leaf litter have different decomposition patterns, and the decomposition rate of mixed leaf litter is faster than that of needle litter in natural state. LN, MN, and HN treatments promoted the decomposition of needle and mixed leaf litter. However, the reduction in soil animal abundance and the different C and N release patterns of needle and mixed leaf litter in HN treatment resulted in the reduction in C:N and the decrease in the decomposition rate of mixed leaf litter. The decomposition constant of litter increased when the two litters were mixed, and the net nutrient transfer between litters tended to occur in low-nutrient species. In general, N addition accelerated the decomposition of litters in *Larix gmelinii* forest, and the addition of *Betula platyphylla* litter also accelerated the decomposition rate of *Larix gmelinii* litter. However, natural atmospheric N deposition contains various inorganic and organic N components. Therefore, a single N addition may not accurately reflect the ecological impacts of atmospheric N deposition on the soil and physicochemical properties of litter decomposition in boreal forest ecosystems. Hence, it is crucial for future research to investigate the effects of different forms of N addition on litter

decomposition and the interactive mechanisms between soil animals and microorganisms for the decomposition of litter.

**Author Contributions:** Conceptualization, Q.W. and M.W.; methodology, Q.W.; formal analysis, Q.W.; investigation, G.L., G.Y., Y.X., Q.W. and M.W.; resources, Q.W.; data curation, G.L., G.Y., Y.X., Q.W. and M.W.; writing—original draft preparation, M.W., Q.W. and G.L.; writing—review and editing, M.W., Q.W., G.Y., Y.X. and G.L.; supervision, Q.W.; funding acquisition, Q.W. All authors have read and agreed to the published version of the manuscript.

**Funding:** This research was supported by grants from the National Natural Science Foundation of China (42230703, 42377477).

**Data Availability Statement:** Data are available from the corresponding author on reasonable request.

**Acknowledgments:** We gratefully acknowledge Ligong Wang from the Academy of Daxinganling Agriculture and Forestry Science, China, for his comments and suggestions regarding an earlier draft of this manuscript.

**Conflicts of Interest:** The authors declare that they have no conflicts of interest.

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
