# Peer review of "Long-Term Nitrogen Addition Accelerates Litter Decomposition in a Larix gmelinii Forest"

_forests, doi:10.3390/f15020372_

Round 1
Reviewer 1 Report
Comments and Suggestions for Authors
Overall, a good article, especially introduction and discussion. But there doesn't seem to be enough novelty. The materials and methods are logically divided into parts, but many little things raise doubts about the correctness of the experiment. Lines 135-136 state that 25 kg of nitrogen is deposited per year, so the nitrogen content cannot be equal to 0 in the control plots! The distance between the plots is only 10m, but it's not enough! Section 2.1. indicates that nitrogen addition was carried out in 2011. And from section 2.2. we learn that the litter bags were laid in 2017, removed in 2018 and 2019. And what happened 6 years after the addition of nitrogen? If the litter bags were laid on 12 sites, then 2 litter bags were removed from the each site each time, and this is an insufficient sample. It is impossible to judge the results, since the figures are unreadable. Table 5 shows data for 3 months, and why not 6 months?
Author Response
Dear Reviewer 1
Thank you very much for your comments concerning our manuscript entitled “Long-term nitrogen addition accelerates litter decomposition in a boreal forest (manuscript ID: forests-2848843)”. This manuscript was completely revised in accordance with your comments. We have resubmitted the manuscript according to your advice, and point-by-point responses to the issues raised by you are as follows.
Q:Overall, a good article, especially introduction and discussion. But there doesn't seem to be enough novelty. The materials and methods are logically divided into parts, but many little things raise doubts about the correctness of the experiment. Lines 135-136 state that 25 kg of nitrogen is deposited per year, so the nitrogen content cannot be equal to 0 in the control plots! The distance between the plots is only 10m, but it's not enough! Section 2.1. indicates that nitrogen addition was carried out in 2011. And from section 2.2. we learn that the litter bags were laid in 2017, removed in 2018 and 2019. And what happened 6 years after the addition of nitrogen? If the litter bags were laid on 12 sites, then 2 litter bags were removed from the each site each time, and this is an insufficient sample. It is impossible to judge the results, since the figures are unreadable. Table 5 shows data for 3 months, and why not 6 months?
A:Thank you for your suggestions. In response to your suggestions, our answers are as follows. (1) The control plot simulates atmospheric nitrogen deposition levels, so no additional nitrogen fertilizer (0 g N ha-1 yr-1) was added, and the local nitrogen deposition level is 25 kg N ha-1 yr-1. (2) Although the distance between sample plots is only 10m, we have used PVC boards to isolate adjacent plots, which can prevent mutual interference between different plots. (3) As you point out that the start of nitrogen addition begin in 2011, and the litter experiment began in 2017, that means litter collected in the litter test started in 2017 has been receiving different doses of exogenous N for 6 years. In order to verify the decomposition process of leaf litter treated with different doses of exogenous nitrogen for 6 years (which could change the carbon to nitrogen ratio of litter), we set up this field experiment in 2017. (4) I am so sorry for my negligence. We have 12 sample plots, with a total of 4 treatments, each repeated 3 times. In every sampling process, 3 needle litter samples and 3 mixed leaf litter samples were taken from each plot, with a total of 72 samples taken one time, and the sum was 432 samples for a total of 6 times during 2 experimental years (2018-2019). (5) Our study focuses more on seasonal dynamics, so we chose to study soil animals in three typical stages of the early (spring), middle (summer), and late (autumn) growth season. Thank you very much again.
We tried our best to improve the manuscript and made some changes in the manuscript. And here we did not list all changes but marked in highlighting in the revised manuscript. Once again, thank you very much for your comments and suggestions.
Yours sincerely,
Qinggui Wang, Ph.D.
Professor
57 Jingxuan West Road,
School of Life Sciences,
Qufu Normal University, Qufu, 273165, China
Tel: +86 537 7038967 Fax: +86 537 7037003
E-mail: [email protected]
Reviewer 2 Report
Comments and Suggestions for Authors
The manuscript by Miao Wang and co-authors represents the interesting results of an experiment lasting two years in the Greater Khingan Mountains. Design of the experiment seems to be fine. And statistics used is adequate.
Major comments:
I suggest adding the region to the title of the manuscript and specifying the type of vegetation e.g., larch forest.
In abstract, I suggest indicating composition of mixed leaf litter (the introduction of Betula platyphylla litter).
I suggest citing in Introduction or in Discussion some of these manuscripts: DOI: 10.3103/s1068373922100077; DOI: 10.1038/s41597-021-00912-z;
Specific comments:
L. 60. ha-1 yr-1
L. 125. 99/ha
L. 147. litters of Larix gmelinii, Betula platyphylla and Larix gmelinii.
L. 146 – 154. I did not catch repetition of bags collected in each period. Please, write in simple words.
L. 146 ‘After air drying’ vs L. 155 ‘dried at 60 ℃ for 48 hours’: it is better to use the same temperature.
L. 163 – 167. Please, add information about pH measurement.
Figures 2 – 5 have inappropriate quality. Please, correct.
L. 244 – 255. Please, shorten. Significant differences? (0.187) > KHN (0.176) > KCK (0.176)
L. 248 decreased by 0.79 year and 1.03. From … to… ? I did not understand.
L. 250. Similar question.
L. 398 15N. superscript.
Comments on the Quality of English LanguageMinor editing of English language is required. I have marked these phrases in the review.
Author Response
Dear Reviewer 2
Thank you very much for your comments concerning our manuscript entitled “Long-term nitrogen addition accelerates litter decomposition in a boreal forest (manuscript ID: forests-2848843)”. This manuscript was completely revised in accordance with your comments. We have resubmitted the manuscript according to your advice, and point-by-point responses to the issues raised by you are as follows.
Major comments:
Q:I suggest adding the region to the title of the manuscript and specifying the type of vegetation e.g., larch forest.
A:Thank you for your suggestion. We have revised the title to "Long-term nitrogen addition accelerates litter decomposition in a Larix gmelinii forest"
Q:In abstract, I suggest indicating composition of mixed leaf litter (the introduction of Betula platyphylla litter).
A: Thank you for your suggestion. We have revised it in lines 12-13.
Q:I suggest citing in Introduction or in Discussion some of these manuscripts: DOI: 10.3103/s1068373922100077; DOI: 10.1038/s41597-021-00912-z;
A: Thank you for your suggestion. We have added references in lines 29 and 355.
[5] Lukina NV, Kuznetsova AI, Geraskina AP, Smirnov VE, Ivanova VN, Teben’kova DN, Gornov AV, Shevchenko NE, Tikhonova EV. Unaccounted Factors Determining Carbon Stocks in Forest Soils. Russian Meteorology and Hydrology. 2022 Oct;47(10):791-803.
[48] Phillips HR, Bach EM, Bartz ML, Bennett JM, Beugnon R, Briones MJ, Brown GG, Ferlian O, Gongalsky KB, Guerra CA, König-Ries B. Global data on earthworm abundance, biomass, diversity and corresponding environmental properties. Scientific data. 2021 May 21;8(1):136.
Specific comments:
Q: L. 60. ha-1 yr-1
A: Thank you for your suggestion. We have revised it in line 59.
Q: L. 125. 99/ha
A: Thank you for your suggestion. We have revised it in line 129.
Q: L. 147. litters of Larix gmelinii, Betula platyphylla and Larix gmelinii.
A: Thank you for your suggestion. We have revised it in lines 152-153.
Q: L. 146 – 154. I did not catch repetition of bags collected in each period. Please, write in simple words.
A: We have 12 sample plots, with a total of 4 treatments, each repeated 3 times. In every sampling process, 3 needle litter samples and 3 mixed leaf litter samples were taken from each plot, with a total of 72 samples taken one time, and the sum was 432 samples for a total of 6 times during 2 experimental years (2018-2019).
Q: L. 146 ‘After air drying’ vs L. 155 ‘dried at 60 ℃ for 48 hours’: it is better to use the same temperature.
A: Thank you for your suggestion. We have revised it in lines 151-153.
Q: L. 163 – 167. Please, add information about pH measurement.
A: Thank you for your suggestion. We have added information about pH measurement in lines 173-175.
Q: Figures 2 – 5 have inappropriate quality. Please, correct.
A: Thank you for your suggestion. We have redrawn Figures 2 – 5.
Q: L. 244 – 255. Please, shorten. Significant differences? (0.187) > KHN (0.176) > KCK (0.176)
A: Thank you for your suggestion. We have revised it in lines 249-251.
Q: L. 248 decreased by 0.79 year and 1.03. From … to… ? I did not understand.
A: Thank you for your suggestion. We have revised it in lines 251-253.
Q: L. 250. Similar question.
A: Thank you for your suggestion. We have revised it in lines 251-253.
Q: L. 398 15N. superscript.
A: Thank you for your suggestion. We have revised it in line 404.
We tried our best to improve the manuscript and made some changes in the manuscript. And here we did not list all changes but marked in highlighting in the revised manuscript. Once again, thank you very much for your comments and suggestions.
Yours sincerely,
Qinggui Wang, Ph.D.
Professor
57 Jingxuan West Road,
School of Life Sciences,
Qufu Normal University, Qufu, 273165, China
Tel: +86 537 7038967 Fax: +86 537 7037003
E-mail: [email protected]
Reviewer 3 Report
Comments and Suggestions for Authors
The paper entitled "Long-term nitrogen addition accelerates litter decomposition in a boreal forest" demonstrated that N addition promoted the decomposition of the litters, inhibited the release of N and P in needle litter and promoted the release of N in mixed leaf litter, highlighting the promoting effect of litter mixing and N addition on litter decomposition.
The manuscript seems to be interesting. However, I have some comments.
Natural atmospheric N deposition contains various inorganic and organic N components. Thus, single N addition may not accurately reflect the ecological effects of atmospheric nitrogen deposition on the physicochemical properties of soil and litter decomposition in forest ecosystems. Therefore, the effects of different forms of N addition on litter decomposition should be investigated.
Some factors like soil enzymes and soil microbial biomass are influenced by N addition and play a key role in litter decomposition in forest ecosystems. It seems that the effects of different forms and mixed forms of N addition on soil microbial biomass and soil-enzyme activities are necessary in litter decomposition. Additionally, some other parameters like humidity index could affect the responses of litter decomposition to N addition.
Litter decomposition is mainly carried out by fungal and bacterial microorganisms. These agents have to be investigated to check their ability in decomposing litter.
The quality of figures must be improved.
Overall, in this study, the mechanism of the promoting effect of litter mixing and N addition on litter decomposition needs to be investigated.
Author Response
Dear Reviewer 3
Thank you very much for your comments concerning our manuscript entitled “Long-term nitrogen addition accelerates litter decomposition in a boreal forest (manuscript ID: forests-2848843)”. This manuscript was completely revised in accordance with your comments. We have resubmitted the manuscript according to your advice, and point-by-point responses to the issues raised by you are as follows.
Q: The paper entitled "Long-term nitrogen addition accelerates litter decomposition in a boreal forest" demonstrated that N addition promoted the decomposition of the litters, inhibited the release of N and P in needle litter and promoted the release of N in mixed leaf litter, highlighting the promoting effect of litter mixing and N addition on litter decomposition. The manuscript seems to be interesting. However, I have some comments.
A: Thanks for your positive comments, and we have made revisions to the entire text according to the reviewer's comments.
Q: Natural atmospheric N deposition contains various inorganic and organic N components. Thus, single N addition may not accurately reflect the ecological effects of atmospheric nitrogen deposition on the physicochemical properties of soil and litter decomposition in forest ecosystems. Therefore, the effects of different forms of N addition on litter decomposition should be investigated.
A: Thank you for your suggestion. In this study, we have chosen ammonium nitrate as the main nitrogen source for our research. In future studies, we will consider the effects of different forms of nitrogen addition on litter decomposition.
Q: Some factors like soil enzymes and soil microbial biomass are influenced by N addition and play a key role in litter decomposition in forest ecosystems. It seems that the effects of different forms and mixed forms of N addition on soil microbial biomass and soil-enzyme activities are necessary in litter decomposition. Additionally, some other parameters like humidity index could affect the responses of litter decomposition to N addition.
A: Thank you for your suggestions. Yes, soil enzymes, microbial biomass, and soil moisture play a crucial role in litter decomposition, and we will consider these factors in future research.
Q: Litter decomposition is mainly carried out by fungal and bacterial microorganisms. These agents have to be investigated to check their ability in decomposing litter.
A: Thank you for your suggestions. Yes, fungi and bacterial microorganisms play a crucial role in the decomposition of litter, and we will take these factors into consideration in future research.
Q: The quality of figures must be improved.
A: Thank you for your suggestion. We have redrawn Figures 2 – 5.
We tried our best to improve the manuscript and made some changes in the manuscript. And here we did not list all changes but marked in highlighting in the revised manuscript. Once again, thank you very much for your comments and suggestions.
Yours sincerely,
Qinggui Wang, Ph.D.
Professor
57 Jingxuan West Road,
School of Life Sciences,
Qufu Normal University, Qufu, 273165, China
Tel: +86 537 7038967 Fax: +86 537 7037003
E-mail: [email protected]
Round 2
Reviewer 1 Report
Comments and Suggestions for Authors
Dear authors, thank you for the explanations and clarifications provided in the article.
Author Response
Dear Reviewer 1
Thank you very much for your comments concerning our manuscript entitled “Long-term nitrogen addition accelerates litter decomposition in a boreal forest (manuscript ID: forests-2848843)”. This manuscript was completely revised in accordance with your comments. We have resubmitted the manuscript according to your advice, and point-by-point responses to the issues raised by you are as follows.
Reviewer #1:
Q:Dear authors, thank you for the explanations and clarifications provided in the article.
A: Thanks for your positive comments, we have made further improvements to the manuscript.
Thank you very much again, and the New Year of the Dragon is coming soon, I wish you a happy Chinese New Year and all the best.
Yours sincerely,
Qinggui Wang, Ph.D.
Professor
57 Jingxuan West Road,
School of Life Sciences,
Qufu Normal University, Qufu, 273165, China
Tel: +86 537 7038967 Fax: +86 5377037003
E-mail: [email protected]
Reviewer 3 Report
Comments and Suggestions for Authors
The authors did not address the questions and just mentioned that they will conduct in the future.
Author Response
Dear Reviewer 3,
Thank you very much for your comments concerning our manuscript entitled “Long-term nitrogen addition accelerates litter decomposition in a boreal forest (manuscript ID: forests-2848843)”. This manuscript was completely revised in accordance with your comments. We have resubmitted the manuscript according to your advice, and point-by-point responses to the issues raised by you are as follows.
Reviewer #3:
Q: The authors did not address the questions and just mentioned that they will conduct in the future.
A: Thank you for your suggestion. We will take these factors into consideration in future research, and we have also supplemented them in the conclusion section in lines 442-448.
Thank you very much again, and the New Year of the Dragon is coming soon, I wish you a happy Chinese New Year and all the best.
Yours sincerely,
Qinggui Wang, Ph.D.
Professor
57 Jingxuan West Road,
School of Life Sciences,
Qufu Normal University, Qufu, 273165, China
Tel: +86 537 7038967 Fax: +86 5377037003
E-mail: [email protected]